# ONEREWARD: UNIFIED MASK-GUIDED IMAGE GENERATION VIA MULTI-TASK HUMAN PREFERENCE LEARNING

## ABSTRACT

In this paper, we introduce OneReward, a unified reinforcement learning framework that enhances the model's generative capabilities across multiple tasks under different evaluation criteria using only *One Reward* model. By employing a single vision-language model (VLM) as the generative reward model, which can distinguish the winner and loser for a given task and a given evaluation criterion, it can be effectively applied to multi-task generation models, particularly in contexts with varied data and diverse task objectives. We utilize OneReward for mask-guided image generation, which can be further divided into several sub-tasks such as image fill, image extend and object removal, involving a binary mask as the edit area. Although these domain-specific tasks share same conditioning paradigm, they differ significantly in underlying data distributions and evaluation metrics. Existing methods often rely on task-specific supervised fine-tuning (SFT), which limits generalization and training efficiency. Building on OneReward, we develop a mask-guided generation model trained via multi-task reinforcement learning directly on a pre-trained base model, eliminating the need for task-specific SFT. Experimental results demonstrate that our unified edit model consistently outperforms both commercial and open-source competitors, such as Ideogram, Adobe Photoshop, and FLUX Fill [Pro], across multiple evaluation dimensions.

## 1 INTRODUCTION

Recent advancements in diffusion model (Rombach et al., 2022; Labs, 2024) have enabled a diverse range of challenging tasks, such as inpainting, outpainting and object removal. Although these tasks share a common mask-guided input format, they exhibit significant divergence in conditional distributions and evaluation metrics, presenting considerable challenge to the development of a unified, versatile model. Inpainting, also known as image fill, involves modifying or adding specific objects within a localized masked area. Outpainting, or image-extend, requires generating extensive content around an existing image, expanding beyond its original borders. Object removal entails filling a masked region based on surrounding context, requires avoiding the generation of extra object, and ensuring texture consistency with the original image. Current state-of-the-art generative models typically excel within specific editing tasks but struggle to maintain consistently high performance across multiple tasks simultaneously. Existing methods or community models often rely on task-specific supervised fine-tuning (SFT), or LoRA (Hu et al., 2022) with limited data based on SD1.5-Inpaint (Rombach et al., 2022) and FLUX Fill (Labs, 2024), which restricts their generalization to diverse editing scenarios. This reveals the difficulty of designing a unified framework capable of supporting multiple image editing tasks while avoiding the inefficiencies of task-specific fine-tuning.

Reinforcement learning from human feedback (RLHF) methods for diffusion and flow matching model, such as Direct Preference Optimization (DPO) (Rafailov et al., 2023; Wallace et al., 2024; Xu et al., 2024; Liu et al., 2025b), reward-base method (Xu et al., 2023; Zhang et al., 2024; Li et al., 2024; Gao et al., 2025) and RL-based method (Black et al., 2023; Liu et al., 2025a; Xue et al., 2025) have shown strong promise in aligning generative outputs with human preferences across text-to-image and text-to-video domains. However, DPO faces fundamental limitations in handling diverse tasks and evaluation dimensions concurrently, as it inherently assumes a well-defined preference

order that may not hold across heterogeneous tasks and criteria. For instance, DPO cannot unambiguously determine the winner and loser when an image is better in aesthetics but worse in structure than its counterpart. Reward Feedback Learning (ReFL), while significantly boosting model performance in specific dimensions, typically requires training separate reward models for each evaluation criterion when using traditional multimodal architectures such as BLIP (Li et al., 2022) and CLIP (Radford et al., 2021), increasing training and tuning complexity. Furthermore, ReFL encounters reward conflicts in multi-task scenarios, where high quality object generation may receive completely opposite evaluations in the task of image-fill and object-removal. FlowGRPO (Liu et al., 2025a) and DanceGRPO (Xue et al., 2025) introduce GRPO (Shao et al., 2024), which is powerful in Large Language Model(LLM), into flow matching models, by converting deterministic Ordinary Differential Equation(ODE) sampleing into a Stochastic Differential Equation (SDE) framework. While GRPO-based methods significantly enhance performance on vision generation tasks, they rely on policy-based estimation by introducing a group-relative formulation to estimate the advantage, without explicitly maximizing reward signals during optimization. This often results in slower convergence compared to reward-driven approaches.

To overcome these limitations, we introduce OneReward, a unified reinforcement learning framework for mulit-task image generation using only one VLM as the reward model. By incorporating task category and evaluation metric information (e.g. aesthetics, structure, consistency) directly into its queries, the VLM can effectively distinguish between tasks and evaluation criteria, enabling it to make pairwise judgments and determine which output is better under certain setting. Base on OneReward, we develop a state-of-the-art (SOTA) mask-guided image generation model that consistently delivers superior performance across a diverse set of tasks, including image fill, image extension and object removal. It is directly optimized via reinforcement learning from a pre-trained model, without any SFT. During training, we treat the initial pre-trained model as the reference model and the training one as the policy model, optimizing the latter to generate results that surpass the reference model in each task-specific evaluation metric. The reward signal is derived from the probability of the token "Yes" generated by the VLM, which is then used for gradient backward. To the best of our knowledge, this is the first work to employ reinforcement learning as a direct optimization paradigm in the context of multi-task image editing. The main contributions of our work are threefold:

1. We propose OneReward, a novel reward model framework for the visual domain by employing VLM as the generative reward model to enhance multi-task reinforcement learning, significantly improving the policy model's generation ability across diverse scenarios.

2. Building on OneReward, we develop a unified SOTA image editing model capable of effectively handling diverse tasks including image fill, image extend and object removal. It surpasses several leading commercial and open-source models, including Ideogram, Adobe Photoshop, and FLUX Fill [Pro].

3. By applying our multi-task reinforcement learning approach on FLUX Fill [dev], we introduce and open-source FLUX Fill [dev][OneReward], a generalized image editing model that outperforms the original model on both inpainting and outpainting tasks, serving as a powerful new baseline for future research in unified mask-guided image generation.

## 2 RELATED WORK

**Mask-guided image generation**: Image inpainting and outpainting focus on generating coherent and seamless content for missing or external regions of an image. Native inpainting variant of Stable Diffusion (Rombach et al., 2022; Podell et al., 2023; Lugmayr et al., 2022) concatenated the latent of the mask and the original image as input to origin text-to-image model. MagicBrush (Zhang et al., 2023) and Inst-Inpaint (Yildirim et al., 2023), introduced refined instruction-based datasets to improve the accuracy of image editing. ByteEdit (Ren et al., 2024) used feedback learning to boost performance in these tasks but implemented task-specific SFT and RL. Recently, FLUX Fill (Labs, 2024) has emerged as a powerful open source baseline demonstrating strong performance in both inpainting and outpainting. However, these models are often specialized or lack robust generalization across multiple, distinct editing modalities. Our unified edit model builds directly upon these foundations, but addresses their limitations by leveraging a novel multi-task RLHF framework, unify inpaint, outpaint, object removal, and text render within a single, proficient model.

**RLHF for diffusion model**: Aligning generative models with human preferences has emerged as a rapidly advancing research area. ReFL (Xu et al., 2023) directly fine-tuned diffusion models based on reward scores and backpropagating them to randomly selected denoise timesteps. Furthermore, VisionReward (Xu et al., 2024) decomposed human preferences into interpretable axes. Without explicit reward model, Diffusion-DPO (Wallace et al., 2024) directly optimize diffusion model based on the preference data. Denoising Diffusion Policy Optimization (DDPO) (Su et al., 2024) was a pioneering work that successfully applied policy gradient methods to diffusion models by casting the denoising process as a multi-step decision-making problem. Further algorithmic advances include Group Relative Policy Optimization (GRPO) (Shao et al., 2024), which has demonstrated strong performance in aligning both diffusion and flow matching models, as evidenced by its applications in FlowGRPO (Liu et al., 2025a), DanceGRPO (Xue et al., 2025), MixGRPO (Li et al., 2025) and Pref-GRPO (Wang et al., 2025). OneReward uses only one VLM as the generative reward model for multi-task reinforcement learning, solving a key limitation of traditional algorithms (e.g., DPO) that fail to distinguish winners/losers under dimension-varying preferences.

## 3 PRELIMINARIES

### 3.1 FLOW MATCHING

Flow Matching (Lipman et al., 2022) represents a new class of generative models that offers a more efficient and stable training paradigm than traditional diffusion models. Instead of learning the score function of a data distribution, flow matching models learn a velocity vector field that transports a simple prior distribution to a complex data distribution through a continuous normalizing flow (CNF). A CNF is defined by an ordinary differential equation (ODE) that describes the trajectory of a sample $\mathbf{x}$ over a continuous time variable $t \in [0, 1]$: $\frac{d\mathbf{x}_t}{dt} = v_t(\mathbf{x}_t)$. Flow matching aims to train a neural network $v_\theta(x, t, c)$ (where $c$ represents conditioning information) to approximate a target vector field $u_t(x|c)$. The conditional flow matching loss is a simple regression objective:

$$\mathcal{L}_{\text{FM}}(\theta) = \mathbb{E}_{t, p_t(\mathbf{x}|\mathbf{c}), \mathbf{c}} \left[ \|v_\theta(\mathbf{x}, t, \mathbf{c}) - u_t(\mathbf{x}|\mathbf{c})\|^2 \right]$$

Rectified Flow (Liu et al., 2022) is a powerful special case of flow matching that linearizes transport paths to maximize sampling efficiency. It simplifies the objective by defining the target vector field as the constant direction between a data sample $x_1$ and its corresponding noise sample $x_0$, such that $u_t(x|c) = x_1 - x_0$. This formulation avoids the complex score-matching objective of diffusion models, which often leads to faster convergence and more efficient generation. Our model is trained upon this efficient training methodology.

### 3.2 REINFORCEMENT LEARNING FROM HUMAN FEEDBACK

RLHF is a powerful technique for aligning generative models with complex, hard-to-specify human preferences. The human preference dataset consists of tuples $(c, x^w, x^l)$, where for each input $c$, $x^w$ is the image preferred by annotators over the alternative output $x^l$. This pairwise preference dataset is used to train a reward model $r_\phi(c, x)$ that predicts a scalar score reflecting human preference. The reward model is often trained using a binary cross-entropy loss based on the Bradley-Terry (Bradley & Terry, 1952) model, which states that the probability of preferring $x^w$ over $x^l$ is:

$$P(x^w \succ x^l | c) = \sigma(r_\phi(x^w, c) - r_\phi(x^l, c))$$

where $\sigma$ is the sigmoid function.

ReFL (Xu et al., 2023) proposes a direct gradient-based fine-tuning method using scalar rewards from a pre-trained reward model. Unlike language models, diffusion models lack a tractable likelihood for complete samples, making traditional RL-based optimization less straightforward to apply. ReFL circumvents this by leveraging the insight that partially denoised samples at latter timesteps already exhibit distinguishable reward scores. During training, ReFL randomly selects a late denoising step $t$, predicts the corresponding image $x_0'$, and computes a scalar reward $r(c, x_0')$ from the reward model. The gradient is then backpropagated through the denoising step using a truncated reward:

$$\mathcal{L}_{\text{reward}} = \text{ReLU}\left(r(c, \hat{x}_0)\right)$$

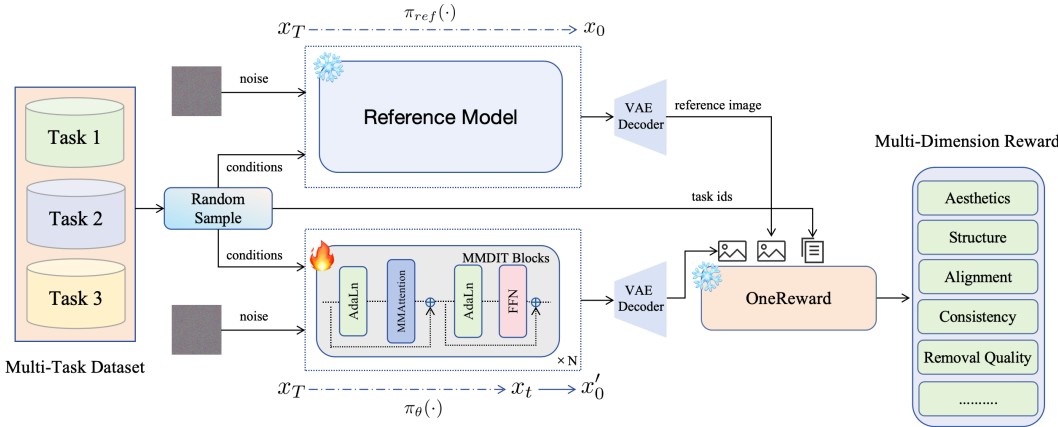

Figure 1: Overall pipeline of our unified RL procedure. We first random sample image and conditions from different task with a certain probability. Start with same condition and different init noise, the reference image is fully denoised using the reference model $\pi_{ref}(\cdot)$. While the evaluation image is partially denoised with randomly selected step and directly predict $x_0'$ based on the policy model $\pi_\theta(\cdot)$. The reward model guides learning by encouraging the policy model to achieve superior performance to the reference model across all evaluation dimensions and tasks.

By applying feedback directly on generation trajectories, ReFL enables preference-driven finetuning of diffusion models in a scalable and model-agnostic manner.

## 4 METHODOLOGY

### 4.1 OVERVIEW

As shown in Fig. 1, we introduce a unified framework for multi-task learning, leveraging the reward model $r$ trained on human preference data to fine-tune the policy model $\pi_\theta$ across several downstream tasks. Let the dataset $\mathcal{D}$ be a collection of $K$ task-specific subsets $\{D_k\}_{k=1}^K$. Each subset $\mathcal{D}_k$ share a common format $\{(x_i, c_i, s_k, \mathcal{E}_k)\}$, where $(x_i, c_i)$ is an image-condition pair, $s_k$ is the task identifier, and $\mathcal{E}_k$ is the set of task specific evaluation metrics. Given current task ids $s_k$ and a certain evaluation metric $e \in \mathcal{E}_k$, the reward model $r$ will be used to calculate the probability how the evaluation image $x_\theta$ is better than reference image $x_{ref}$, which are generated by policy model $\pi_\theta$ and reference model $\pi_{ref}$, respectively. In the process of our RL pipeline, the objective of our training scheme is to increase the probability of generated reward-aligned tokens, as determined by the reward model $r$. OneReward enables the model to efficiently learn a versatile, multi-task generation policy that satisfies multidimensional human preferences within a single, unified process.

This section is structured as follows: we first detail our data construction pipeline in Section 4.2. Next, in Section 4.3, we describe the training procedure for our reward model, termed OneReward. Finally, we present the multi-task, multicriteria reinforcement learning strategy in Section 4.4 and Section 4.5.

### 4.2 HUMAN PREFERENCE DATA COLLECTION

To support the development and evaluation of our unified image editing framework, we construct a large-scale, high-quality human preference dataset for multi-task image generation. This dataset spans four major editing tasks: image fill, image extend, object removal, and text rendering, each presenting unique challenges and requiring distinct evaluation perspectives.

In image fill and image extend, the model is required to synthesize plausible content in user-specified regions, guided by natural language prompts. In contrast, the object removal task centers on the elimination of specified elements from the input image. To maintain a unified input format across all tasks, we assign a universal, content-agnostic prompt (e.g., "remove") to all object removal instances. Each sample is structured as a triplet $(I_{src}, M, P)$, where $I_{src} \in \mathbb{R}^{H \times W \times 3}$ is the source

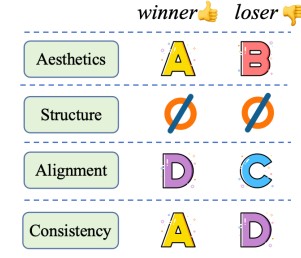

Prompt: A close-up portrait of a child with freckles
and red hair, wearing a light-colored top.

Figure 2: Illustration of the pairwise annotation process. Given multiple candidate outputs for the same prompt and binary mask, annotators identify the best and worst samples under each evaluation dimension to form a winner/loser pair. If the differences between candidates are negligible, the dimension is discarded (denoted by $\emptyset$), ensuring that only informative comparisons are retained. To clarify, this showcase uses an all-one mask, meaning the entire image region is generated.

image, $M \in \mathbb{R}^{H \times W}$ is the binary mask indicating the region to be edited, and $P$ is the corresponding text prompt. This unified input representation enables consistent processing across different tasks within our framework. Our data collection pipeline utilizes a pre-trained diffusion model and intentionally randomizes its key inference parameters to generate a diverse set of candidate images for each input sample.

For multi-task and multi-dimensional evaluation, we design a task-specific annotation protocol including structural validity, internal consistency, image-text alignment, aesthetics, and removal quality. For image fill and image extend, each set of candidate images is evaluated along the first four dimensions. And for object removal, where no user text is present, we reduce the evaluation to the final single criterion. Our data annotation process, illustrated in Fig. 2, is based on a Best-of-$N$ and Worst-of-$N$ selection scheme. For each input triplet $(I_{src}, M, P)$ and its $N$ candidate outputs, annotators identify the most and least preferred image. This selection is performed independently for every evaluation dimension, yielding a dataset of preference tuples that enables fine-grained, dimension-aware preference supervision.

By collecting preference pairs at the metric level, rather than through holistic or averaged scoring, our annotation strategy overcomes a major limitation of prior work by enabling the disentanglement of conflicting judgments across distinct evaluation dimensions. This comparative and multi-dimensional evaluation scheme promote high-quality and robust human perference data, which serves as a strong foundation for training reward models in multi-objective image generation tasks.

## 4.3 Multi-Dimensional Pairwise Reward Model Training

Designing a unified multi-task reward model for image editing faces two primary challenges: scalability across diverse tasks and robustness against reward hacking. Conventional scalar rewards often fail in mask-guided tasks by focusing on unchanged backgrounds rather than the critical edited region, which results in unreliable signals that poorly represent the quality of intended edit.

To address these limitations, we propose OneReward, a unified reward model designed to assess outputs across multiple image editing tasks and evaluation dimensions with only one reward model. OneReward is built upon the perceptual capabilities of a pre-trained VLM, which serves as the backbone for all-dimension reward prediction. Rather than introducing task-specific output heads, we guide the model through a textual evaluation query $q$ , which encodes both the task identifier $s_k$ and the evaluation dimension $e \in \mathcal{E}_k$. This formulation enables dynamic conditioning of the VLM on the specific aspect of quality to be evaluated. Formally, the evaluation query is constructed as:

$$q = \Phi(s_k, e, P) \tag{1}$$

where $\Phi$ is a instruction template as shown in Fig. 3, $P$ is the original input prompt, used only when evaluating prompt–image alignment. For all other dimensions, the query remains prompt-free, ensuring that judgments reflect intrinsic visual quality without semantic bias.

$$y^+ \mid y^-$$

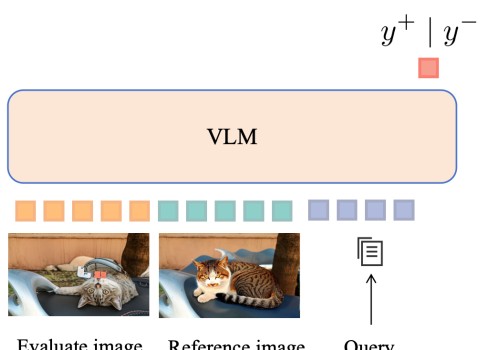

VLM

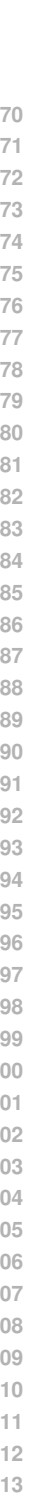

Evaluate image    Reference image    Query

*QUERY:*
*Both of these two images are generated results of [TASK]. Please evaluate the difference of the two images. Is the **first** image better than the **second** one? Please answer "yes" or "no". The evaluation criteria is [DIMENSION].<The prompt of the two image is [PROMPT].>*

Figure 3: The detail of our one reward model. We utilize VLM to judge whether the first image is better than the second one. In the process of reward feedback learning, the probability of $y^+$ token is treated as the reward to the diffusion models. We simply add the edit task and the evaluation dimensions to the user query, achieving the goal of training for different task and dimensions. The content of angle brackets is optional, only add when the evaluation dimension is Text Alignment.

To further enhance robustness and address the shortcomings of absolute scoring, OneReward employs a pairwise comparison evaluation scheme. As illustrated in Fig. 3, the reward model takes a pair of images $(x^w, x^l)$ along with an evaluation query $q$ as input, and outputs a binary classification that aligns with the human preference for the specific query. It is defined as:

$$y = r(x^w, x^l, q) \tag{2}$$

Our training methodology derives directly from the Best-of-N and Worst-of-N annotation protocol in Sec. 4.2. This process allows each preference pair to be labeled with one or more evaluation dimensions. For each training sample, we construct the corresponding evaluation queries $q$ for all the annotated dimensions and compute the standard cross-entropy loss. The loss function is defined as follows:

$$\mathcal{L}(\phi) = -\frac{1}{2}\mathbb{E}_{(x^w, x^l, q)\in\mathcal{D}}\left[\log\mathbb{P}_\phi\left(y^+ \mid x^w,\ x^l,\ q\right) + \log\mathbb{P}_\phi\left(y^- \mid x^l,\ x^w,\ q\right)\right] \tag{3}$$

where $y^+$ denotes affirmative token (e.g. "Yes"), $y^-$ denotes negative token (e.g. "No"), and $\mathbb{P}_\phi$ represents the probability assigned to the corresponding token by the reward model. $x^w$ and $x^l$ refer to the winner and loser pair under a specific evaluation criterion. This training strategy enables the efficient utilization of our annotations, empowering the reward model with the capacity to perform nuanced, multi-dimensional evaluations across a range of tasks.

### 4.4 MULTI-TASK REWARD FEEDBACK LEARNING

Based on OneReward, we propose a novel framework that systematically aligns a pre-trained diffusion model with complex, multi-dimensional human preferences across a suite of tasks. As shown in Fig. 1, the whole train pipeline is composed of three main components: a frozen reference model $\pi_{ref}$, a trainable policy model $\pi_\theta$, and the trained OneReward model $r$. In each iteration, we first sample data from the sub-dataset $\mathcal{D}_k$ according to a predefined sampling probability $p_k$. The reference model $\pi_{ref}$ is initialized by the parameters of the pre-trained diffusion model. It generates a baseline image $x_{ref}$ via a full denoising trajectory from $x_T$ to $x_0$. Inspired by ReFL (Xu et al., 2023), the evaluate image $x_\theta$ starts from the same condition $c$ but with a different noise vector, performing partial denoising up to a randomly selected intermediate timestep $t$ before directly predicting the final latent $x_0'$. The vae decoder is used to reconstruct pixel-space images from these latents and gradients are only through this final single-step prediction. Both the prediction of the evaluate $x_\theta$ and the reference image $x_{ref}$, along with an evaluation query $q$, are passed into the reward model. The reward model evaluates whether $x_\theta$ is preferred over $x_{ref}$ under the given task and evaluation dimension, and returns a binary output token $y^+$ and $y^-$:

$$x_\theta = \text{vae\_decode}(x_0')$$
$$x_{\text{ref}} = \text{vae\_decode}(x_0) \tag{4}$$
$$y = r(x_\theta, x_{\text{ref}}, q)$$

---

**Algorithm 1** Multi-Task Reinforcement Learning from Human Feedback

---

**Dataset:** Multi-Task image-condition datasets $\{\mathcal{D}_k\}_{k=1}^K$, with data sample probability distribution $\mathcal{P} = \{p_1, p_2, \ldots, p_K\}$, task ids $S = \{s_1, s_2, \ldots, s_K\}$ and each evaluation dimension $\{\mathcal{E}_k\}_{k=1}^K$.
**Input:** Reference diffusion model $\pi_{ref}$, policy model $\pi_\theta$ with parameters $\theta$, unified reward model $r$ with parameters $\phi$, hyperparameters $[t_1, t_2]$ for the generation of evaluate image.
 1: Init reference model $\pi_{ref} \leftarrow \pi_\theta$
 2: Init ema model $\pi_{ema} \leftarrow \pi_\theta$
 3: **for** iteration = 1, ..., N **do**
 4:     Sample condition $c$ from the $k$-th dataset $\mathcal{D}_k$ with probability $p_k$
 5:     Sample init noise $\epsilon_1, \epsilon_2$ from normal distribute $\mathcal{N}(0, 1)$
 6:     Random sample denoise timesteps $t$ from $[t_1, t_2]$
 7:     Generate the reference image $x_{ref}$ with full denoise procedure $\pi_{ref}(\epsilon_1, c)$
 8:     Generate the evaluate image $x_\theta$ with random denoise steps $\pi_\theta(\epsilon_2, c, t)$
 9:     **for** $e \in \mathcal{E}_k$ **do**
10:         Generate query $q$ with task id $s_k$ and current evalution dimension $e$ as shown in Fig.3
11:         Compute RL loss $\mathcal{J}_e(x_\theta, x_{ref}, q)$ in Equation5 with reward model $r$
12:     **end for**
13:     Updata policy model via gradient ascent:$\pi_\theta \leftarrow \pi_\theta + \frac{1}{|\mathcal{E}_k|} \nabla_{\pi_\theta} \sum_{e \in \mathcal{E}_k} \mathcal{J}_e$
14:     EMA update $\pi_{ema} \leftarrow \tau \pi_{ema} + (1 - \tau)\pi_\theta$
15: **end for**
**Output:** $\pi_\theta, \pi_{ema}$

---

As $x_\theta$ and $x_{ref}$ are generated from the same condition $c$ and different model parameters, we simplify the rollout procedure as $\pi_\theta(\cdot)$ and $\pi_{ref}(\cdot)$, respectively. During RL, the probability to response $y^+$ is treated as the reward signal. And the objective of the policy model is to maximize this expected reward, the loss function is defined as follows.

$$\mathcal{J}(\theta) = \max\left(0, \lambda - \mathbb{P}_\phi\left(y^+ \mid \pi_\theta(c), \ \pi_{\text{ref}}(c), \ q\right)\right) \tag{5}$$

where $y^+$ denotes the affirmative token and $\mathbb{P}_\phi$ is the probability predicted by the reward model parameterized by $\phi$, $\lambda$ is the predefined reward upper bound, $q$ is the query. Our training objective for the policy model involves simultaneously optimizing for all relevant evaluation metrics of a given task. This encourages the model to achieve balanced improvements across diverse evaluation criteria, rather than overfitting to any single aspect of generation. A detailed description of the multi-task reinforcement learning process can be found in Alg. 1.

### 4.5 DYNAMIC REINFORCEMENT LEARNING

As shown in Alg. 1, our training pipeline typically maintain three models in parallel: a policy model, a reference model, and an EMA variant. This design leads to high memory consumption and increases the engineering complexity of model synchronization. On the other hand, if the reference images are of insufficient quality, the policy model may be trained on overly easy preference pairs, potentially leading to reward hacking and hindering effective learning.

To address these limitations, we propose a dynamic reinforcement learning strategy (Alg. 2), in which the EMA model is directly reused as the reference model. As training progresses, the EMA model gradually improves in generative quality, thereby providing an increasingly strong baseline for policy comparison. This design not only reduces memory overhead by eliminating the need for a separate reference model, but also yields more stable and adaptive reward signals throughout training. A conceptual visualization of the reward computation process is provided in Figure 9, highlighting the difference between the baseline setup and this dynamic design. In the dynamic variant, the reference model is continuously enhanced during training, ensuring that the policy is always compared against a strong and progressively improving baseline.

Table 1: Accuracy(%) of the reward model across multiple editing tasks and evaluation dimensions. Each entry denotes the model's accuracy in distinguishing winners from losers on the test set for a given dimension. Higher values indicate more reliable preference discrimination by the reward model. The first row reports results using the original weights in (Bai et al., 2025) without finetuning. The subsequent rows with Qwen2.5VL* reports results of reward model finetuned on paired data. "+task" indicates that task information is included in the VLM query, while "+dimension" further incorporates the evaluation dimension information.

| Task | Image Fill | | | | Image Extend | | | | Object Removal |
|---|---|---|---|---|---|---|---|---|---|
| Evaluate Dimension | Text Alignment | Consistency | Structure | Aesthetics | Text Alignment | Consistency | Structure | Aesthetics | Removal Quality |
| Qwen2.5VL (no finetune) | 54.45 | 50.60 | 58.71 | 57.71 | 52.25 | 53.33 | 53.95 | 54.26 | 48.44 |
| Qwen2.5VL* | 65.84 | 67.12 | 71.74 | 65.84 | 58.56 | 69.63 | 68.12 | 68.34 | 84.87 |
| Qwen2.5VL*+task | 69.39 | 68.67 | 74.20 | **76.26** | 57.66 | 70.00 | **71.52** | 70.47 | **86.88** |
| Qwen2.5VL*+task+dimension | **81.14** | **74.01** | **74.57** | 74.82 | **73.87** | **82.22** | 71.25 | **72.17** | 86.66 |

Table 2: Performance improvements with different reward models. We apply reinforcement learning to Flux Fill [dev] using different reward models trained with distinct system queries. "—" stands for the baseline of official model or API without RL. Reward-guided optimization consistently boosts performance across PickScore (Kirstain et al., 2023), ImageReward (Xu et al., 2023), HPSv2 (Wu et al., 2023), HPSv3 (Ma et al., 2025), ReMOVE (Chandrasekar et al., 2024). Importantly, the RL-enhanced variants of Flux Fill [dev] substantially outperform not only its base counterpart but also the closed-source Flux Fill [pro].

| Base Model | Reward Model | Image Fill | | | | Image Extend | | | | Object Removal |
|---|---|---|---|---|---|---|---|---|---|---|
| | | PickScore↑ | ImageReward↑ | HPSv2↑ | HPSv3↑ | PickScore↑ | ImageReward↑ | HPSv2↑ | HPSv3↑ | ReMOVE↑ |
| | — | 0.7618 | 0.2364 | 0.2150 | 0.6486 | 0.7398 | 0.0601 | 0.2128 | 0.3769 | 0.7246 |
| | Qwen2.5VL (no finetune) | 0.7724 | 0.6048 | 0.2260 | 0.7677 | 0.7494 | 0.3242 | 0.2292 | 0.5605 | 0.7287 |
| Flux Fill [dev] | Qwen2.5VL* | 0.7776 | 0.6845 | 0.2294 | 0.8203 | 0.7566 | 0.4678 | 0.2344 | 0.6451 | 0.7377 |
| | Qwen2.5VL*+task | 0.7772 | 0.6928 | 0.2286 | 0.8099 | 0.7597 | 0.5163 | 0.2355 | 0.6539 | 0.7338 |
| | Qwen2.5VL*+task+dimension | **0.7778** | **0.7066** | **0.2293** | **0.8285** | **0.7615** | **0.5479** | **0.2361** | **0.6575** | **0.7404** |
| Flux Fill [pro] | — | 0.7723 | 0.5526 | 0.2251 | 0.8080 | 0.7533 | 0.3720 | 0.2275 | 0.5982 | 0.7295 |

## 5 EXPERIMENTS

### 5.1 IMPLEMENTATION DETAILS

**Dataset**. Our dataset for multi-task image editing was constructed through a multi-stage pipeline. We began by extracting image embeddings using the CLIP model (Radford et al., 2021), followed by K-means clustering to obtain 100k diverse and representative subset for reinforcement learning. Based on the selected subset, we further generated and annotated 120k human preference pairs, which serve as training data for both the reward model and the image generation model. For evaluation, we constructed a diverse benchmark consisting of 130 images for image fill, 200 for image extend and 100 for object removal. It covers a wide range of scenes and artistic styles.

**Experimental Settings**. For our reward model, we fine-tuned the open-source Qwen2.5-VL-7B-Instruct (Bai et al., 2025) on our preference dataset with batch size 16, learning rate 1e-6. We adopt our internal model and Flux Fill [dev] as the text-to-image base model respectively. For our internal model, data were sampled with probabilities of 50% for image fill, 25% for image extend, and 25% for object removal. And for Flux Fill [dev] the probabilities are 35%, 35% and 30%. Training was performed with batch size 8, learning rate 5e-6.

**Evaluation Metrics**. We use PickScore (Kirstain et al., 2023), ImageReward (Xu et al., 2023), HPSv2 (Wu et al., 2023), HPSv3 (Ma et al., 2025) and ReMOVE (Chandrasekar et al., 2024) to evaluate the edited results.

### 5.2 RESULTS AND ABLATION STUDY

As shown in Tab. 1 and Tab. 2, we evaluate four types of reward models: (i) Qwen2.5VL (no finetune), which directly uses the pretrained weight in Bai et al. (2025) without any training process; (ii) Qwen2.5VL*, finetuned on paired preference data; (iii) Qwen2.5VL*+task, where task-specific information is added into the query; and (iv) Qwen2.5VL*+task+dimension, our proposed OneReward model that further incorporates evaluation-dimension signals, as shown in Fig. 3. The results in Tab. 1 demonstrate that progressively enriching the reward model with task and dimension knowl-

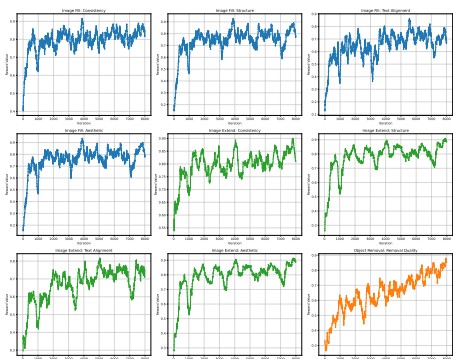 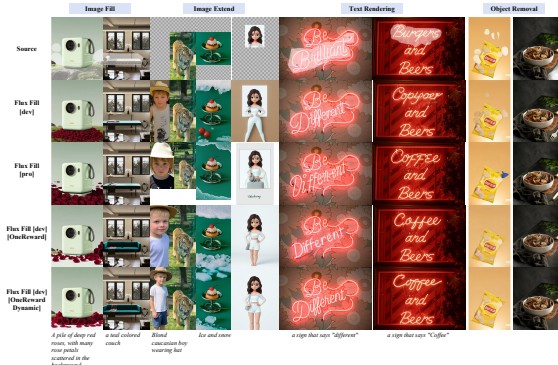

Figure 4: We visualize the reward curves of Consistency, Structure, Text Alignment, Aesthetics for image fill (blue) and image extend (green), and Removal Quality for object removal (orange).

Figure 5: Visual comparison of results for our RL-ed model. Rows correspond to different methods, columns show task-specific prompts and outputs. The source images are shown in the first row. The last two rows stand for our RL-enhanced model, trained via Alg. 1 and Alg. 2.

edge substantially increases its accuracy in distinguishing human preferences across editing tasks. As shown in Tab. 2, reward models with higher accuracy provide stronger and more reliable supervision signals for reinforcement learning, which in turn facilitates more effective optimization. By applying reinforcement learning to Flux Fill [dev], we consistently observe improvements in aesthetics, alignment, and overall quality as the reward model becomes stronger. Notably, with the full task-and-dimension variant, our RL-enhanced Flux Fill [dev] outperform not only its base counterpart but also the closed-source Flux Fill [pro].

Fig. 4 illustrates the reward curves across different evaluation dimensions during training, covering three tasks: image fill (blue), image extend (green), and object removal (orange). Each subplot shows the evolution of reward values over training iterations for specific task–dimension pairs, including Consistency, Structure, Text Alignment, Aesthetics, and Removal Quality. As shown in in this figure, all curves display a clear upward trend, indicating consistent performance gains across dimensions. It suggests that while multi-task optimization inherently introduces greater variability due to shared capacity and competing objectives, the model still converges reliably with steady improvements. The high final reward values achieved across all dimensions further validate the effectiveness of our reward-driven training scheme.

### 5.3 QUALITATIVE RESULTS

Based on Flux Fill [dev], we applied both Alg. 1 and Alg. 2 for RL training. Fig. 5 shows a comparison of the resulting models with Flux Fill [dev] and Flux Fill [pro]. Our method achieves superior structural fidelity and content quality, yielding more coherent inpainting, smoother outpainting, precise text rendering, and artifact-free object removal. It shows that our RL-enhanced model, trained via OneReward, demonstrates superior visual quality compared to the base model and even the closed-source API.

## 6 CONCLUSIONS

We present OneReward, a unified reward model designed for multi-task, multi-dimensional reinforcement learning of diffusion and flow matching model. It enables fine-grained supervision across diverse tasks by leveraging VLM as the reward model. Built on OneReward, our proposed mask-guided generative model, which achieves SOTA performance on image fill, image extend, object removal, and text rendering, outperforming both commercial APIs and open-source models on most dimensions. To further support the community, we open-source FLUX Fill [dev][OneReward], an enhanced variant of flux model trained with our RL framework, offering stronger generalization and broader task applicability.

## ETHICS STATEMENT

Our work builds on the open-source FLUX Fill [dev] and Qwen2.5-VL-7B-Instruct, which is publicly available. All data utilized in our study has undergone a thorough legal and ethical review to ensure compliance with privacy regulations and data protection principles. Our methodology and the application of our model do not raise any new ethical concerns and we are dedicated to the safe and principled advancement of generative AI technologies.

## REPRODUCIBILITY STATEMENT

The core methodology of our work, including the data construction, reward model training, and the multi-task reward feedback learning, is described in detail in the paper. Our proposed method can be reproduced by following the described steps in algorithm 1 and algorithm 2. To facilitate full reproduction and to encourage further research, we will open-source the final model together with supporting materials.

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

# A  APPENDIX

## A.1  THE USE OF LARGE LANGUAGE MODELS(LLMS)

Large Language Models (LLMs) were only used to correct grammar errors and polish the writing. They were not involved in research ideation, experiment design, analysis, or other substantive contributions.

## A.2  COMPARISONS WITH STATE OF THE ARTS

We employ OneReward on our internal text-to-image model, and conducted extensive comparisons with SOTA models and APIs, including Ideogram, Higgsfield, Adobe Photoshop, Midjourney, and Flux Fill [Pro]. We conducted a user study involving 40 participants. Each participant rated the generated images across multiple dimensions: overall quality, text alignment, texture consistency, style consistency, structural plausibility, aesthetics, text rendering, and removal quality. Among these, overall usability, text rendering, and removal quality were treated as binary judgments, where each image was assessed as either acceptable or not. The reported values for these metrics therefore represent success rates expressed as percentages. In contrast, the remaining dimensions were rated on a 1–5 Likert scale, and the scores were averaged to produce Mean Opinion Scores (MOS), where higher values indicate better quality.

Comparative results for the four image editing tasks are presented in Fig. 6 and Tab. 3. Our unified edit model demonstrates the strongest overall performance across all tasks. For image fill, our model achieves a usability rate of 69.04%, outperforming the second-best competitor (52.11%) by 16.93 percentage points. It also obtains the highest scores in most dimensions, including text alignment, texture consistency, structure, aesthetics, and text rendering, with the only exception being style consistency, where Ideogram shows a slight advantage. On the text-guided image extend task, our model performs comparably to Ideogram while clearly surpassing Flux Fill [pro] and Adobe Photoshop in usability. And in the text-free setting, our model shows pronounced superiority, achieving the highest usability rate (87.54%) and leading across all reported dimensions. For object removal, our model again delivers the best results, with a usability rate of 82.22% and a removal quality score of 86.33%, significantly outperforming other SOTA competitors. The high removal quality indicates that our model produces the fewest unwanted objects in this task, behavior that typically conflicts with goals in other generation tasks such as image fill or extend. This demonstrates the effectiveness of our RL strategy in multi-task human preference learning.

## A.3  ABLATION STUDY

To assess the impact of OneReward, we conduct a Good–Same–Bad (GSB) evaluation that compares our generation model trained with and without reward guidance. As shown in Fig. 7, each bar represents the distribution of human preferences among different tasks. Compared to the base model, the OneReward variant receives a higher proportion of "Good" ratings in all tasks. The GSB results demonstrate that our unified reward model generally shifts model output toward preferred generations.

To further validate the effectiveness of the query-guided mechanism in OneReward, we conducted an ablation study on the trained reward model under two distinct settings, as reported in Tab. 4. In the first setting, we adopt task- and dimension-specific queries as illustrated in Fig. 3. In the second setting, we instead use a single fixed query, where all test pairs are judged under the task of Object Removal with the evaluation dimension of Removal Quality. The results clearly indicate that using the fixed query leads to a significant drop in accuracy across all tasks and dimensions compared to using specific queries. This performance degradation provides strong evidence that our model has successfully learned to disentangle and apply different evaluation criteria based on the provided query, rather than relying on an undifferentiated judgment pattern.

## A.4  QUALITATIVE RESULTS

Comparison results are shown in Fig. 8. Our method consistently excels across diverse tasks. For image fill, it generates structurally sound and stylistically consistent objects that adhere to the prompt.

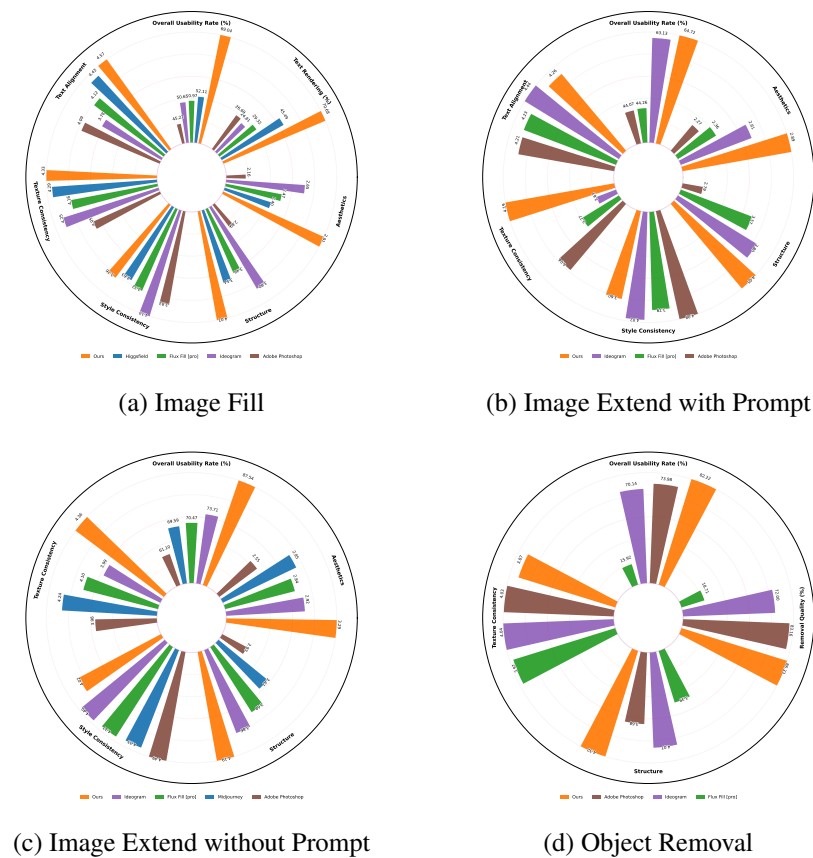

(a) Image Fill

(b) Image Extend with Prompt

(c) Image Extend without Prompt

(d) Object Removal

Figure 6: Overall evaluation across four image editing tasks, and text rendering is included in image fill. For each sub-task, we selected only state-of-the-art models or closed-source APIs as competitors and conducted detailed evaluations in multiple dimensions. Note that different tasks have different evaluation criteria.

As shown in the first column, only our method correctly renders the specified "black iphone". In the second column, competing models show clear failures: Ideogram fails to generate the "tiger"; the tiger from Flux Fill[pro] is stylistically inconsistent with the background; and the one from Photoshop suffers from severe structural artifacts in its limbs. In image extend, our approach again shows the best consistency and structural quality. In the fourth column's examples, only our model plausibly completes the girl's missing limbs, which other methods failed to accomplish. For Text Rendering, our model achieves high textual accuracy, ensuring the generated text precisely matches the input prompt. For Object Removal, our method yields the cleanest and most seamless results, outperforming all competitors.

And representative qualitative examples of our model are provided in Fig. 10. These examples, featuring images of diverse visual styles, are selected to highlight two key capabilities of our model: its ability to generate high-quality edits and its robustness in adapting to various stylistic contexts.

## A.5 COMPARISONS WITH ORIGIN QWEN2.5VL(NO FINETUNE)

Figure 11 provides a compelling visual comparison between the outputs from (a) the original Flux-fill[dev], (b) a model trained with the original Qwen2.5VL as the reward model, and (c) a model trained with our fine-tuned OneReward. It is evident that our model produces objects with better structural integrity than the one trained with the original Qwen2.5VL. Moreover, the original Qwen2.5VL does not focus on background integration, resulting in images with overly bright colors (Row 4) and clear segmentation between generated content and original content (Row 7). Addition-

Table 3: Quantitative comparison of our model against SOTA competitors across four editing tasks. Metrics with percentages (e.g., Usability Rate, Text Rendering, Removal Quality) are reported as success rates, while other dimensions (e.g., Text Alignment, Texture Consistency, Style Consistency, Structure, Aesthetics) are Mean Opinion Scores (MOS) rated on a 1–5 scale. Higher scores indicate better performance in all dimensions. Our model demonstrates consistently superior performance across most dimensions compared with existing SOTA models, especially in overall usability rate.

| Task | Model | Usability Rate(%) | Text Alignment | Texture Consistency | Style Consistency | Structure | Aesthetics | Text Rendering(%) | Removal Quality(%) |
|---|---|---|---|---|---|---|---|---|---|
| Image Fill | Adobe Photoshop | 45.22 | 4.09 | 4.05 | 3.83 | 2.89 | 2.16 | 26.69 | – |
| | Ideogram | 50.65 | 3.78 | 4.25 | **4.13** | 3.80 | 2.65 | 24.81 | – |
| | Flux Fill [pro] | 50.97 | 4.12 | 4.16 | 3.72 | 3.46 | 2.47 | 29.32 | – |
| | Higgsfield | 52.11 | 4.43 | 4.29 | 3.63 | 3.54 | 2.40 | 45.49 | – |
| | **Ours** | **69.04** | **4.57** | **4.33** | 3.76 | **4.02** | **2.91** | **70.68** | – |
| Image Extend w Prompt | Adobe Photoshop | 44.07 | 4.21 | 4.04 | **4.08** | 2.79 | 2.27 | – | – |
| | Flux Fill [pro] | 44.26 | 4.23 | 3.77 | 3.79 | 3.57 | 2.36 | – | – |
| | Ideogram | 63.13 | **4.44** | 3.63 | 4.02 | 3.80 | 2.61 | – | – |
| | **Ours** | **64.72** | 4.26 | **4.19** | 3.60 | **4.05** | **2.89** | – | – |
| Image Extend w/o Prompt | Adobe Photoshop | 61.10 | – | 3.98 | **4.49** | 2.93 | 2.55 | – | – |
| | Midjourney | 69.59 | – | 4.24 | 4.33 | 3.47 | 2.95 | – | – |
| | Flux Fill [pro] | 70.47 | – | 4.10 | 4.37 | 3.68 | 2.84 | – | – |
| | Ideogram | 73.71 | – | 3.99 | 4.40 | 3.86 | 2.92 | – | – |
| | **Ours** | **87.54** | – | **4.36** | 4.02 | **4.19** | **3.29** | – | – |
| Object Removal | Flux Fill [pro] | 15.92 | – | 3.97 | – | 3.39 | – | – | 18.71 |
| | Ideogram | 70.14 | – | **4.04** | – | 4.07 | – | – | 72.00 |
| | Adobe Photoshop | 73.98 | – | 4.03 | – | 3.68 | – | – | 83.16 |
| | **Ours** | **82.22** | – | 3.87 | – | **4.32** | – | – | **86.33** |

Table 4: Accuracy(%) of OneReward model across multiple editing tasks and evaluation dimensions. In the first row, we provide the model with task- and dimension-specific queries. While in the second row, the query is fixed to the task of Object Removal with the evaluation dimension of Removal Quality.

| Task | Image Fill | | | | Image Extend | | | | Object Removal |
|---|---|---|---|---|---|---|---|---|---|
| Evaluate Dimension | Text Alignment | Consistency | Structure | Aesthetics | Text Alignment | Consistency | Structure | Aesthetics | Removal Quality |
| Multi Queries | **81.14** | **74.01** | **74.57** | **74.82** | **73.87** | **82.22** | **71.25** | **72.17** | **86.66** |
| Fixed Query | 43.15 | 65.75 | 53.24 | 56.35 | 45.95 | 72.22 | 58.86 | 59.12 | 86.66 |

ally, the original Qwen2.5VL does not understand the removal task, as the generated images still contain debris, failing to complete the removal of the object.

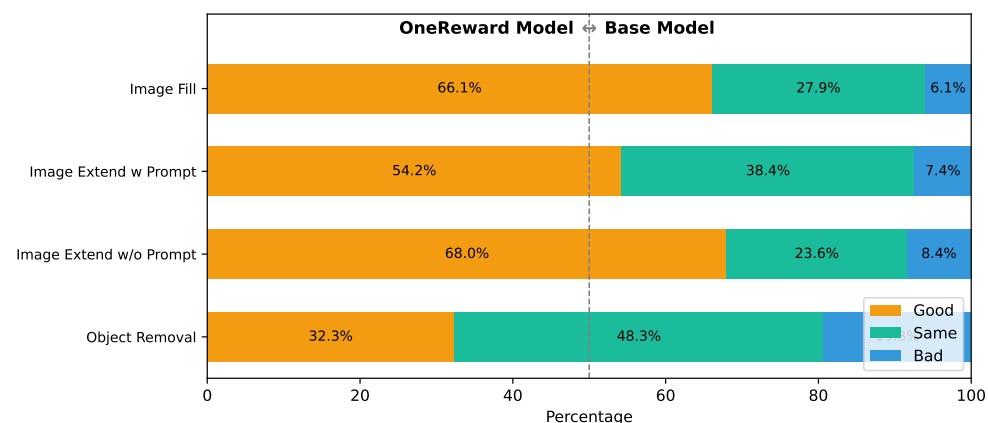

Figure 7: Comparison of performance between our RL-ed model and base model using Good–Same–Bad (GSB) evaluation. Each group corresponds to a specific image editing task. Bars represent the relative proportions of outputs judged as Good (orange), Same (green), or Bad (blue) across different model pairs. This visualization highlights the distribution of relative preferences, showing where OneReward-enhanced models outperform the base model.

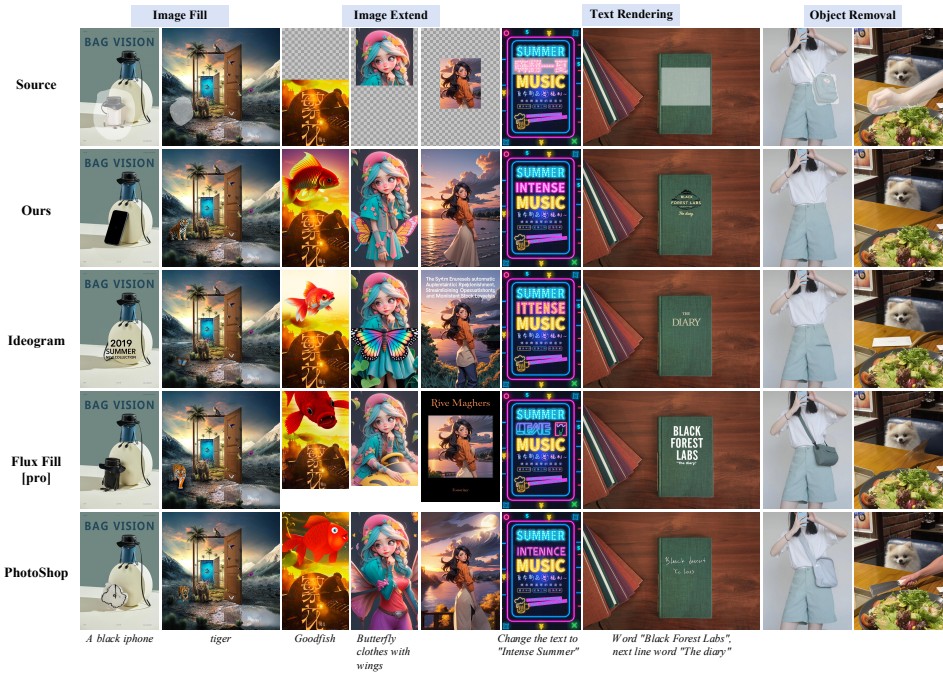

Figure 8: Visual comparison of editing results for our unified edit model and its competitors across different tasks. Rows correspond to different methods, columns show task-specific prompts and outputs. The source images are shown in the first row. The blank row at the bottom indicates that the case is prompt-free.

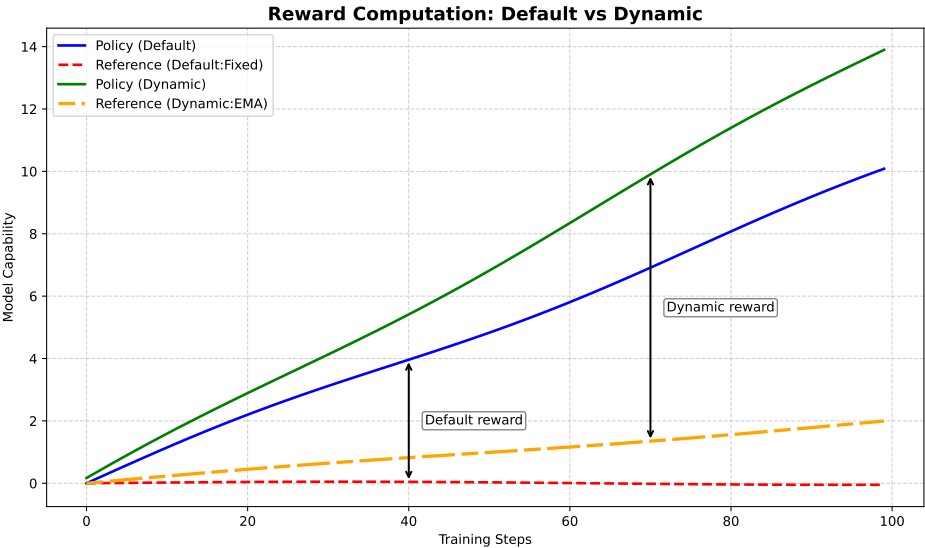

Figure 9: Schematic illustration of reward computation in the baseline and our dynamic framework. In the baseline 1, rewards are measured as the vertical gap between the policy (blue) and a fixed reference (red). In the dynamic method 2, rewards are computed against an EMA-updated reference (orange) that evolves smoothly with the policy (green), forming a dynamic baseline. This figure is for conceptual understanding only and does not reflect actual parameter values or training dynamics.

---

**Algorithm 2** Dynamic Multi-Task Reinforcement Learning from Human Feedback

---

**Dataset:** Multi-Task image-condition datasets $\{\mathcal{D}_k\}_{k=1}^K$, with data sample probability distribution $\mathcal{P} = \{p_1, p_2, \ldots, p_K\}$, task ids $S = \{s_1, s_2, \ldots, s_K\}$ and each evaluation dimension $\{\mathcal{E}_k\}_{k=1}^K$.
**Input:** Reference diffusion model $\pi_{ref}$, policy model $\pi_\theta$ with parameters $\theta$, unified reward model $r$ with parameters $\phi$, hyperparameters $[t_1, t_2]$ for the generation of evaluate image.
  1: Init reference model $\pi_{ref} \leftarrow \pi_\theta$
  2: **for** iteration = 1, ..., N **do**
  3:     Sample condition $c$ from the $k$-th dataset $\mathcal{D}_k$ with probability $p_k$
  4:     Sample init noise $\epsilon_1, \epsilon_2$ from normal distribute $\mathcal{N}(0, 1)$
  5:     Random sample denoise timesteps $t$ from $[t_1, t_2]$
  6:     Generate the reference image $x_{ref}$ with full denoise procedure $\pi_{ref}(\epsilon_1, c)$
  7:     Generate the evaluate image $x_\theta$ with random denoise steps $\pi_\theta(\epsilon_2, c, t)$
  8:     **for** $e \in \mathcal{E}_k$ **do**
  9:         Generate query $q$ with task id $s_k$ and current evalution dimension $e$ as shown in Fig. 3
 10:         Compute RL loss $\mathcal{J}_e(x_\theta, x_{ref}, q)$ in Equation 5 with reward model $r$
 11:     **end for**
 12:     Updata policy model via gradient ascent: $\pi_\theta \leftarrow \pi_\theta + \frac{1}{|\mathcal{E}_k|} \nabla_{\pi_\theta} \sum_{e \in \mathcal{E}_k} \mathcal{J}_e$

 13:     ▷ EMA update $\pi_{ref} \leftarrow \tau \pi_{ref} + (1 - \tau)\pi_\theta$
 14: **end for**
**Output:** $\pi_\theta, \pi_{ref}$

---

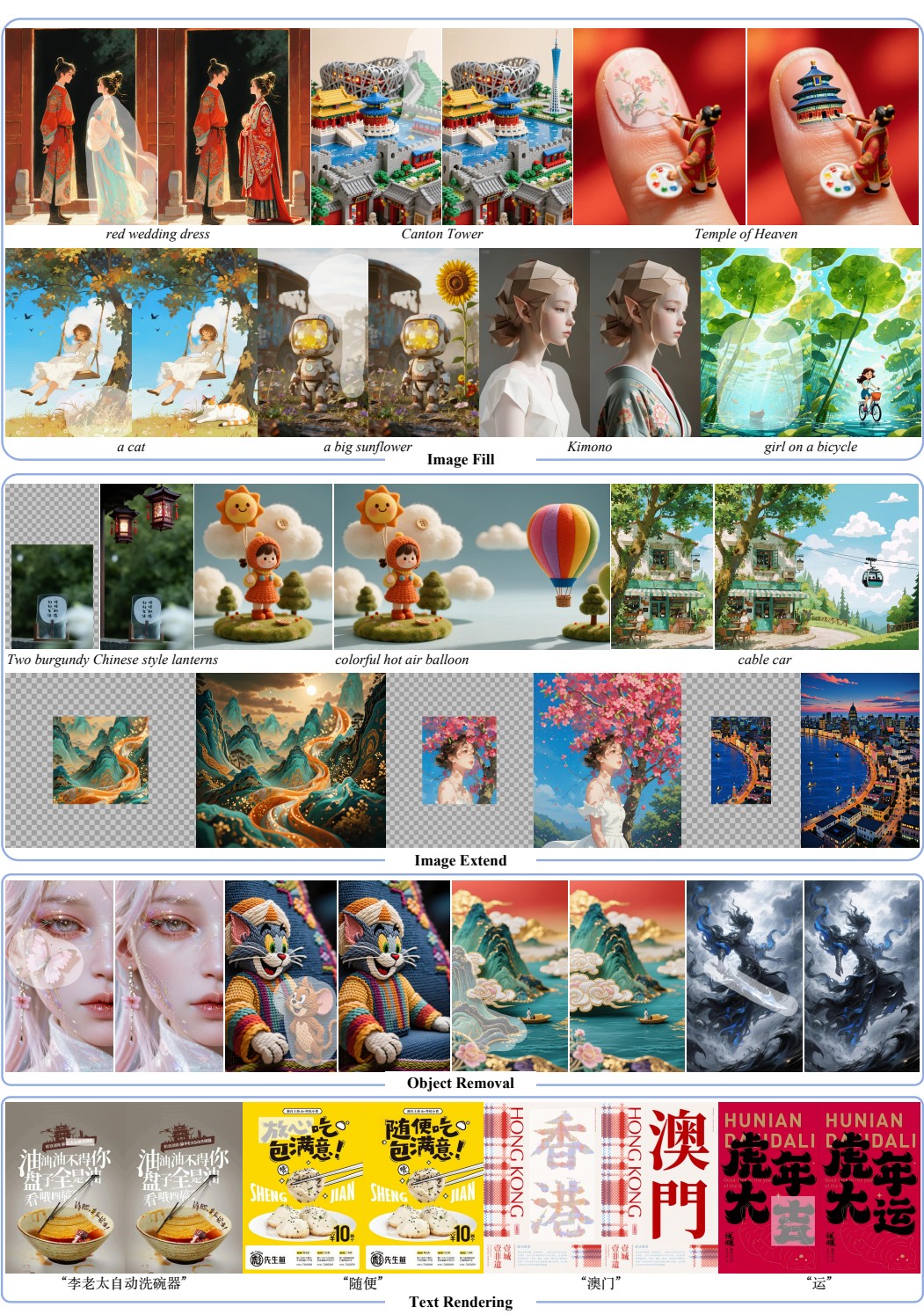

Figure 10: Visual showcase of our model across four scenario: image fill, image extend, object removal and text rendering. Each column presents a representative example with corresponding prompts and outputs, demonstrating the model's unified capability across diverse generation objectives.

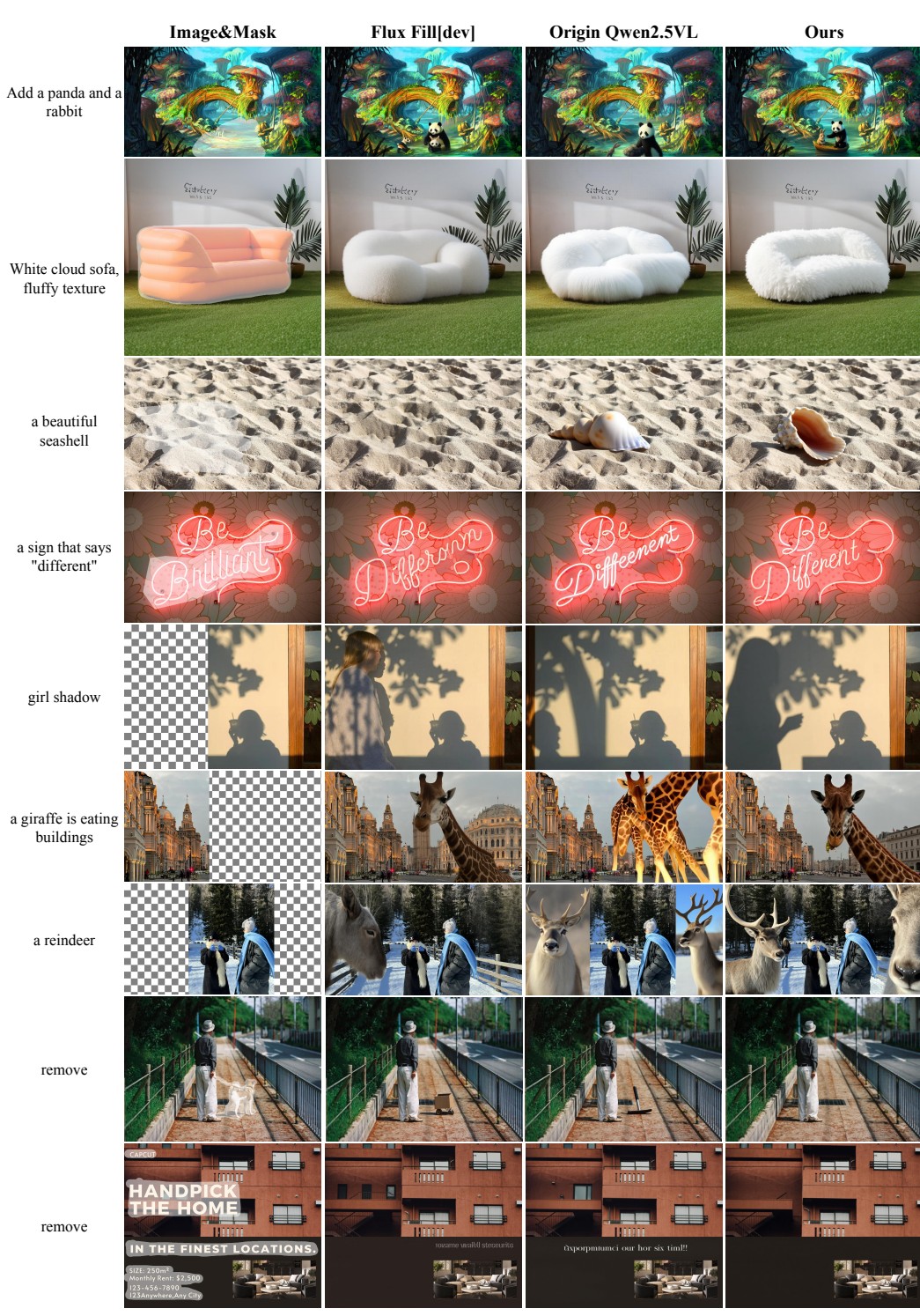

Figure 11: Visual comparison of editing results for origin Flux Fill[dev], model trained by Qwen2.5VL (no finetune) and our unified edit model across different tasks. Columns correspond to different methods, rows show task-specific prompts and outputs. The source images are shown in the first row.

