# OpenReview forum: "OneReward: Unified Mask-Guided Image Generation via Multi-Task Human Preference Learning"
_ICLR.cc/2026/Conference — Submitted to ICLR 2026_

### Official Review · Reviewer_vrmd · 2025-10-26

**Soundness:** 2
**Presentation:** 3
**Contribution:** 3
**Rating:** 4
**Confidence:** 3

**Summary:**

This paper introduces OneReward, proposing a unified reward model that are fine-tuned with preference data to achieve accurate assessment for reinforcement learning. The main contributions are two-fold: (1) a novel pipeline that considers task ID as a condition to achieve reward unification. (2) reinforcement learning pipeline with the proposed reward model to achieve image editing on various tasks (image inpainitng, outpaining, etc).  Extensive experiments demonstrate their methods' efficacy.

**Strengths:**

(1) OneReward aims to unify reward based on the given task, which is beneficial to achieve compact image editing training.

(2) The overall writting is good and method is easy to understand.

**Weaknesses:**

(1) One of the main contributions for this work is training a unified reward model to achieve comprehensive evaluation for reinforcement learning. However, improving generation quality could also be achieved by jointly using multiple reward experts. The paper lacks comparison or discussion with the ``mixture-of-reward-experts'' variant in Tab.2, which is proven to be effective in reinforcement finetuning [a].

(2) In Tab.1, recent VLMs such as [b] is also capable of comparing two images based on the given instruction. It is necessary to compare your uni-reward model with them.

(3) I am curious about the method's  training efficiency. Specifically, the reward is assigned on the final generated results, which means a full-time sampling before obtaining a reward. The whole process is time-consuming, especially when being deployed on FLUX-based  models.

[a] T2I-R1: Reinforcing Image Generation with Collaborative Semantic-level and Token-level CoT. In NIPS'25.
[b] Q-Insight: Understanding Image Quality via Visual Reinforcement Learning. In NIPS'25.

**Questions:**

Please refer to "weaknesses"

---

> ### Author Response · Authors · 2025-11-23
>
> ## Response to Weakness 1
> Thank you for your insightful comment. As you pointed out, several prior works adopt a mixture-of-reward-experts strategy, where different reward models are used for different tasks or evaluation dimensions. For example, [a] employs multiple reward experts including human preference model, object detector, VQA model and output reward model. Each reward model is responsible for a specific evaluation aspect. **The core idea of using multiple reward models is to obtain multi-dimensional assessment signals.**
>
> ### 1. Experiment Results
> However, we observe that **a single Vision-Language Model (VLM) processes strong image-understanding and discriminative capabilities, sufficient to evaluate multiple sub-tasks across multiple dimensions**. The high accuracy of our unified reward model across multiple tasks and dimensions, as detailed in Table 1 of our paper, provides strong evidence for this claim and is one of our key contributions.
>
> We also trained a separate reward model for each individual evaluation dimension, and their accuracies are reported below (the second row). Actually, the accuracy of dimension-specific reward models is nearly identical to that of our unified reward model. This indicates that a single VLM is indeed capable of performing reliable multi-dimensional evaluations without the need for multiple specialized reward experts.
>
> | Tasks | Image Fill | |  |  | Image Extend | |  |  | Object Removal |
> |----------|----------|----------|----------|----------|----------|----------|----------|----------|----------|
> | Dimensions | Text Alignment | Consistency | Structure | Aesthetics | Text Alignment | Consistency | Structure | Aesthetics | Removal Quality |
> | Unified Qwen2.5VL | 81.14 | 74.01 | 74.57 | 74.82 | 73.87 | 82.22 | 71.25 | 72.17 | 86.66 |
> | Separate Qwen2.5VL | 82.92 | 73.57 | 74.90 | 74.58 | 74.65 | 80.36 | 71.61 | 73.42 | 85.61 |
>
> Our findings suggest that VLMs have sufficiently strong perceptual and reasoning capabilities to function as a unified multi-task reward model. This not only reduces the complexity and computational cost of multi-task reinforcement learning, but also avoids the burden of tuning and maintaining multiple reward models during training.
>
> ---
>
> ## Response to Weakness 2
> Thank you for the valuable suggestion.
> ### 1. Differences in Evaluation Metrics
> While [b] indeed proposes using VLMs for image quality assessment, their model focuses on **low-level image degradations** such as noise, darkening, or other distortion types. However, mask-guided image editing tasks require evaluation along different dimensions, including structural correctness of generated objects, aesthetic quality, text-image alignment, the consistency between generated content and the background, and the cleanliness of object removal. Therefore, the evaluation criteria of [b] are not well aligned with the requirements of our image editing tasks, making a direct comparison less meaningful.
>
> ### 2. Gradient Inaccessibility
> Furthermore, our approach is based on **Refl (Reinforcement Learning from Human Feedback)**, a reward-based method that requires **backpropagating gradients** from the reward signal to update the generator. To achieve this, we use the probability of a fixed token (e.g. "Yes") output from the reward model as a differentiable reward signal. However, [b] employs a Chain-of-Thought reasoning process, which involves generating free-form text through multiple next-toke prediction steps. It is difficult to get gradients from the complex multi-step textual outputs. As a result, the reward model in [b] is not directly compatible with Refl training.
>
> ---
>
> ## Response to Weakness 3
> During the training of the generation model, we ran 8000 steps, which took about **70 hours** in total. Instead of using the full 50 sampling steps of FLUX, we use only **20 steps** to generate the reference image and **10-15 steps** to produce the policy image. In practice, low-step sampling is sufficient to provide effective reward signals for RL training and still leads to substantial performance improvements. Prior work FlowGRPO[c] also reduces the timesteps of data collection to enhance the training speed.
>
> [c]  Liu, J., Liu, G., Liang, J., Li, Y., Liu, J., Wang, X., ... & Ouyang, W. (2025). Flow-grpo: Training flow matching models via online rl. arXiv preprint arXiv:2505.05470.

---

### Official Review · Reviewer_Np2y · 2025-10-31

**Soundness:** 3
**Presentation:** 3
**Contribution:** 3
**Rating:** 6
**Confidence:** 4

**Summary:**

The paper introduces "OneReward", a unified reinforcement learning framework for enhancing generative models, particularly in mask-guided image generation tasks. By employing a single VLM as the reward model, the framework allows the model to efficiently distinguish between the winner and loser in multi-task settings with varied data distributions and evaluation metrics. The proposed model aims to eliminate task-specific fine-tuning (SFT) and offers a unified approach for tasks such as image inpainting, object removal, image extension, and text rendering. Experimental results demonstrate that OneReward outperforms both commercial and open-source competitors across several metrics. The model offers improvements over existing generative models, enhancing flexibility, scalability, and performance.

**Strengths:**

The introduction of OneReward as a unified reward model for multiple image editing tasks is highly innovative, addressing a clear gap in current image generation models that rely on task-specific fine-tuning. The paper provides strong experimental evidence showing that OneReward outperforms several competitive methods across multiple tasks. The ability to apply OneReward to diverse tasks like image fill, object removal, and text rendering without needing task-specific fine-tuning is a significant strength. This work could significantly advance the field of multi-task generative models, serving as a new baseline for future research.

**Weaknesses:**

The multi-task reinforcement learning framework is complex and could be challenging to train and fine-tune, especially on large-scale datasets. The paper could have delved deeper into potential limitations of the framework, particularly with respect to edge cases where the model might struggle with certain tasks or where performance might degrade. The training time and computational cost could also be discussed further, as multi-task reinforcement learning can be computationally expensive. While the experimental results are convincing, a more detailed analysis of the failure cases or scenarios where OneReward may not perform as expected would be useful.

**Questions:**

1.Can the OneReward framework be adapted to other domains beyond image editing, such as video generation or more complex multimodal tasks?
2.What are the specific computational costs associated with training the OneReward model, and how does this compare to task-specific models in terms of efficiency?
3.In multi-task reinforcement learning, how do you ensure the model doesn't overfit to one task at the expense of others? More details on the regularization methods used to balance task performance would be helpful.

---

> ### Author Response · Authors · 2025-11-23
>
> ## Response to Question 1
> Yes, absolutely. The OneReward framework is designed as a general training methodology for improving model performance across multiple tasks and evaluation dimensions simultaneously. As long as human preference data is available for a target domain, the framework can be directly adapted to video generation or complex multimodel tasks.
>
> Moreover, we believe the approach is highly promising. As the field moves towards larger, multi-purpose foundation models, a multi-task, multi-dimensional reinforcement learning framework like OneReward is particularly well suited, as it provides a unified and scalable way to optimize diverse behaviors across tasks.
>
> ---
>
> ## Response to Question 2
> ### 1. Training Time and Computational Cost
> Our training process consists of two stages. **Training the reward model** for 5 epochs required approximately 22 hours. **Training the generative model** for 8000 steps took around 70 hours. During the ReFL stage, although the full Flux inference typically uses 50 steps, we only use **20 steps** for generating reference images and 10-15 steps for producing policy images. In practice, low-step inference remains effective for RL training (as also noted in Flow-GRPO[a]).
> ### 2. Compared with Task-specific Training
> Regarding efficiency, previous task-specific training requires separate supervised fine-tuning (SFT) and individual reinforcement learning (RL) for each task. In contrast, our approach performs **unified multi-task RL** directly on a general base model. We eliminate the need for task-specific supervised fine-tuning and avoiding per-task RL training. This greatly reduces overall training time and computational cost while still improving model performance across tasks.
>
> [a] Liu, J., Liu, G., Liang, J., Li, Y., Liu, J., Wang, X., ... & Ouyang, W. (2025). Flow-grpo: Training flow matching models via online rl. arXiv preprint arXiv:2505.05470.
>
> ---
>
> ## Response to Question 3
>  We prevent inter-task overfitting and balance performance primarily through two mechanisms: **task-aware reward signal** and **difficulty-based data sampling**.
> ### 1. Task-aware Reward
> First, we explicitly encode the task information and evaluation dimension into the input instruction of the reward model. As shown in Table 1 of the main paper, the reward model achieves high accuracy across all sub-tasks in the test set. This validates the reward model is able to correctly assess the performance of the generative model across different sub-tasks, which provide an unbiased reward signal for multi-task RL.
> ### 2. Difficulty-based Data Sampling
> Second, in the multi-task reinforcement learning stage, we use a weighted sampling strategy that allocates more training data to more difficult tasks. This prevents the model from quickly mastering and overfitting to simpler tasks while neglecting more complex ones. As illustrated by the reward curves in Figure 4, the model improves across all sub-tasks during training, empirically demonstrates the efficiency of this sampling strategy.

---

### Official Review · Reviewer_UTa8 · 2025-11-01

**Soundness:** 1
**Presentation:** 2
**Contribution:** 3
**Rating:** 4
**Confidence:** 4

**Summary:**

This work proposes a unified reinforcement learning framework that uses a single vision-language model as a generative reward model to supervise multi-task image generation under varied evaluation criteria. Applied to mask-guided editing (image fill, extension, object removal, text rendering), it replaces task-specific SFT with multitask RL on a pretrained base model, despite differing data distributions and metrics. Experiments show the unified edit model consistently outperforms commercial and open-source systems across multiple evaluation dimensions.

**Strengths:**

- The idea of leveraging powerful VLMs as reward models is promising.

- Internal model shows promising results, even outperforming commercial models.

- The authors propose a high-quality preference model and dataset, which will be open-sourced.

**Weaknesses:**

Thanks to the authors’ efforts in this manuscript and their commitment to open-sourcing the data and model to benefit the community. However, I have several concerns about the work.

- Because VLMs such as Qwen-2.5-VL are heavily pre-trained, they should be treated as general reward models. The substantial jump from baselines to the *no-finetune* setting in Table 2 supports this. However, the proposed fine-tuning yields only marginal gains over *no-finetune*, which substantially weakens its contribution to the preference dataset.

- In the open-source setting, key fair comparisons with RLHF baselines (e.g., DanceGRPO, MixGRPO, Pref-GRPO) are missing.

- Because the main qualitative results omit text rendering, some claims appear overstated (e.g., in the Abstract). The logical organization should be revised.

- Lacks comparison with general-purpose image editing models, such as MagicBrush, Qwen-Image-Edit, and Gemini-Banana.

- The ablation studies are insufficient. For example, the impact of probability-based data sampling is not analyzed, and the per-task contributions are unclear. Given the authors’ statement that "different sub-tasks differ significantly in underlying data distributions and evaluation metrics," this raises concerns about potential task conflicts, which may misguide the community’s research direction.

**Questions:**

The high-level motivation is to leverage VLMs as reward models to guide generation. Why not compare against unified understanding-and-generation models?

**Details Of Ethics Concerns:**

While I appreciate the authors’ efforts to open‑source the dataset very much, the introduction of a new image dataset raises concerns about human privacy and potential bias.

---

> ### Author Response · Authors · 2025-11-23
>
> ## Response to Weakness 1
> Thank you for your insightful comment.
>
> ### 1. Qwen2.5VL’s Capabilities and Limitations
>
> As you pointed out, Qwen2.5VL has undergone extensive pre-training on large datasets, equipping it with powerful image understanding and assessment. During training, the inference steps for the policy image (10-15 steps) are fewer than those for the reference image (20 steps), which results in noticeable differences in image quality and sharpness. The general knowledge of the base Qwen2.5VL allows it to correctly identify this quality disparity. As a result, origin Qwen2.5VL can gradually improve the generation quality of the policy image during the Refl phase.
>
> This effect is quantitatively reflected in Table 2. The metrics for image-fill and image-extend primarily assess image quality and aesthetics, where Qwen2.5VL contributes improvements. However, for object removal, which assesses the cleaniness of the area, Qwen2.5VL shows only marginal improvement (FluxFill[dev]: 0.7246 → Qwen2.5VL: 0.7287 → Ours: 0.7404). While the base VLM can optimize for generic quality, it fails to grasp specific task objectives.
>
> ### 2. Task-Specific Requirements Beyond Image Quality
> However, simply improving image quality and clarity does not meet the requirements of our three image editing tasks. We aim to enhance the structural accuracy of the generated content, the seamless integration with the background, text-to-image alignment, the visual details and aesthetics of the generation, and the cleaniness of object removal.
>
> ### 3. Visual Comparison
> To further illustrate the performance differences, we add a visual comparison of the results from Flux-fill[dev], origin Qwen2.5VL and our proposed model to the Appendix (Figure 11).  It is evident that our model produces objects with better structural integrity than the one trained with the original Qwen2.5VL. Moreover, the original Qwen2.5VL does not focus on background integration, resulting in images with overly bright colors (Row 4) and clear segmentation between generated content and original content (Row 7). Additionally, the original Qwen2.5VL does not understand the removal task, as the generated images still contain debris, failing to complete the removal of the object. More images are shown in supplementary materials.
>
> In summary, while the base Qwen2.5VL provides a generic quality signal, it is insufficient for the nuanced requirements of our three image editing tasks.
>
> ---
>
> ## Response to Weakness 2
> We thank the reviewer for their valuable feedback. We would like to further clarify **the core contributions of our work and its distinction from the GRPO family of methods**.
>
> First, the core contribution of our work lies in **the design of the reward model**. We leverage a VLM to perform ranking-based comparisons between two input images, which effectively mitigates the influence of non-edited regions in image editing tasks and provides higher robustness compared to single-image scoring approaches. In addition, we successfully developed a unified reward model that can handle multiple tasks and multiple evaluation dimensions.
>
> Secondly, our training paradigm is based on **Refl (Reinforcement Learning from Human Feedback)**, which is a reward-based optimization paradigm that aims to maximize the reward assigned to the diffusion model’s outputs. In contrast, GRPO methods are policy-gradient based and optimize the denoising trajectory. These two approaches are fundamentally different in both their optimization objectives and training mechanism. Moreover, methods such as DanceGRPO and MixGRPO rely on single-image scoring, whereas our proposed OneReward uses pairwise ranking between images, making these approaches not directly applicable.
>
> ---
>
> ## Response to Weakness 3
>  We thank the reviewer for pointing this out. In the main paper, we have grouped the text rendering task under the image-filling category. We will revise the description in the abstract to ensure consistent terminology regarding text rendering.

---

> ### Author Response · Authors · 2025-11-23
>
> ## Response to Weakness 4 & Question 1
> We thank the reviewer for this question. The primary focus of our work is mask-guided local image editing, where the core objective is to **modify only the specified region while ensuring the non-masked areas remain strictly unchanged**.
>
> This constitutes a fundamental distinction from recent **instruction-based editing models**, such as MagicBrush, Qwen-Image-Edit, and Gemini-Banana, as well as from unified understanding-and-generation models like Qwen-Image-Edit and OmniGen2. These models operate **without an input mask** and, crucially, do not prioritize or guarantee the preservation of unedited regions. Our approach, by contrast, is specifically designed for high-fidelity, localized control.
>
> Looking ahead, we plan to develop a future version of our model capable of general-purpose, instruction-based editing. This will involve building upon **powerful foundational models** such as FLUX.1-Kontext and Qwen-Image-Edit.
>
> ---
>
> ## Response to Weakness 5
> We thank the reviewer for raising concerns regarding probability-based data sampling and multi-task learning.
>
> ### 1. Probability-based Data Sampling
> First, the sampling ratios used in our experiments are indeed empirical parameters obtained through observation. The motivation is that **more challenging tasks require a higher sampling frequency during training**. From the standpoint of task difficulty, we find that image filling > image extending > object removal. Object removal uses a fixed text prompt and requires lower content diversity. Image extending uses a rectangular mask with limited variation. In contrast, image filling involves both complex masks and textual prompts, making it the most challenging.
>
> Consequently, in our internal model, we allocate 50% to image filling, and 25% each to image extending and object removal. For Flux-Fill[dev], it has already been supervised-fintuned on image filling and image extending. Its initial performance on object removal is weaker. Therefore, we increase the sampling ratio of object removal to 30%.
>
> ### 2. Multi-task Learning
> Second, we would like to clarify that we do not claim that sub-tasks conflict with each other. The key issue is that **different sub-tasks have different objectives and evaluation criteria**. For instance, in mask-guided image generation, image filling aims to generate specific content consistent with input textual description, while object removal aims to seamlessly fill the masked region without introducing new objects. Using a single reward model such as ImageReward or HPSv2 would only evaluate aesthetic or perceptual quality, and cannot distinguish between these **task-specific criteria**. By explicitly injecting both the task information and evaluation dimensions into the VLM’s instruction input, our method enables **fine-grained and task-aware evaluation**, allowing the reward model to judge each sub-task according to the appropriate criteria.
>
> ---
>
> ## Response to Ethics Concerns
> Before releasing the dataset, it will undergo a full legal review, and all content containing potential privacy-related information will be strictly filtered out.

---

### Official Review · Reviewer_PA5S · 2025-11-11

**Soundness:** 3
**Presentation:** 3
**Contribution:** 3
**Rating:** 4
**Confidence:** 3

**Summary:**

This paper introduces OneReward, a smart framework that uses a single, powerful VLM as a flexible "judge." By telling this one reward model what task and what metric to evaluate, it can be used to train a single generator for multiple, conflicting tasks. This unified model ends up beating specialized SOTA competitors like Adobe Photoshop and FLUX Fill.

**Strengths:**

- Using a single, promptable VLM as a multi-task, multi-metric judge is a brilliant, scalable idea.

- They successfully trained one model to do conflicting tasks and win.

- It outperforms strong commercial and open-source competitors in head-to-head comparisons.

**Weaknesses:**

- The entire system relies 100% on the VLM (Qwen2.5-VL) being an accurate and unbiased judge.

- It's unclear how a single generator learns to be good at opposite goals simultaneously.

- They had to build a massive, 120k-pair multi-dimensional preference dataset, which is very hard to reproduce.

**Questions:**

How does the single generator model handle the conflicting objectives? Does getting better at "object removal" ever make it worse at "image fill"?

---

> ### Author Response · Authors · 2025-11-23
>
> ## Response to Weakness 1
> As the reviewer points out, our training framework relies on a single VLM judge. We address the concern in three aspects:
> ### 1. Experimental Results
> Firstly, we conduct an **experiment to validate the accuracy of the VLM's judgements**. As shown in Table 1 of the main paper, we evaluated our fine-tuned Qwen2.5-VL on a test set with human preference labels. This set spans nine evaluation criteria across three image edit tasks (image filling, image extending and object removal). The experimental results show that Qwen2.5-VL achieves an accuracy of over 70% across all dimensions, and surpasses 80% on several. This provides strong evidence that our model effectively learns human preferences and produces judgments highly consistent with human annotations, supporting it is a reliable reward model.
> ### 2. Approach in the Current Study
> Secondly, traditional approaches (e.g. ImageReward, HPSv2, PickScore) typically rely on CLIP/BLIP/VGG-based architectures. With the recent progress of large visual-language models, **pioneering studies (e.g., VisionReward, VideoReward, UnifiedReward)** have shifted towards leveraging VLMs as reward models. Due to their strong general knowledge and superior image understanding capabilities, VLMs can provide more robust and reliable image judgments. Our work builds upon this by extending the VLM's capability into a multi-task multi-dimensional reward framework, which constitutes a core contribution of our research.
> ### 3. Multi-dimensional Reward Design
> Our reward formulation integrates **multiple independent evaluation dimensions**, including aesthetics, structure, consistency, text-image alignment and removal cleaniness. Rather than relying on a single score, the multi-dimensional reward ensures a more holistic optimization and mitigates the risk of training being dominated by a single dimension.
>
> ---
>
> ## Response to Weakness 2 & Question
> We appreciate the reviewer's insightful question regarding how a single generator can effectively handle multiple tasks. Our framework achieves this through **task-specific conditioning for both the generator and reward model**.
> ### 1. Task-specific Condition for Generator
> The diffusion model fundamentally learns **the conditional probability $p(x|c)$**. Different image editing tasks correspond to different forms of condition $c$ with diverse distribution and characteristics. For example, in the object removal task, we use a fixed predefined text prompt as the condition; in the image extending task, the goal is to extend the image outward along its height and width, which corresponds to a fixed rectangular mask; and in the image filling task, we use randomly shaped masks together with textual inputs. These **task-specific condition inputs** guide the generator to perform the corresponding editing behavior.
> ### 2. Task-specific Reward
> In addition, we explicitly incorporate task information into the reward model. We predefined distinct sets of evaluation criteria for each task and embedded this information into the reward model. During training, the reward model provides targeted feedback by selectively applying task-relevant evaluation dimensions. This mechanism enables the generator to improve its performance under different conditions by receiving specific guidance for each task.
> ### 3. Experimental Results
> Fig. 4 in the main paper illustrates the **reward curves** for all evaluation dimensions across the three image editing tasks, showing that our generator model achieves consistent performance improvements across tasks during training.

---

> ### Author Response · Authors · 2025-11-23
>
> ## Response to Weakness 3
> We sincerely thank the reviewer for raising this crucial point regarding the data requirements for training.
> 1. We agree with the reviewer that current reward models usually require a substantial amount of human preference data to ensure that VLMs can accurately learn true human preferences. This trend is also reflected in prior work: ImageReward[a] constructed 130k human preference image pairs, VideoReward[b] collected 182k human preference video annotations, and VisionReward[c] included 97k human preference image labels. These works consistently show that building a sufficiently large and high-quality preference dataset is crucial for training a reliable reward model. However, in the field of image editing, there is currently no large-scale and high-quality human preference dataset available.
> 2. In our experiments, we observed that the model can converge after roughly 8000 iterations using 8×H20 GPUs. Based on this empirical observation, we estimate that around 64k human preference samples are sufficient for training. To ensure data diversity, we first performed clustering over a million images and selected 100k representative samples. We then constructed diverse image editing tasks from these samples and collected human preference annotations, resulting in a final high-quality dataset of 120k samples.
> 3. We recognize that the reliance on large-scale manual annotation is a practical bottleneck for the broader adoption and scaling of such models. Addressing this is a key focus of our planned future research. We will further explore using a small amount of human-annotated data to efficiently fine-tune a VLM, and then leveraging the VLM’s judgment capability to automatically construct large-scale synthetic preference datasets.
>
> [a] Xu, J., Liu, X., Wu, Y., Tong, Y., Li, Q., Ding, M., ... & Dong, Y. (2023). Imagereward: Learning and evaluating human preferences for text-to-image generation. Advances in Neural Information Processing Systems, 36, 15903-15935.
>
> [b] Liu, J., Liu, G., Liang, J., Yuan, Z., Liu, X., Zheng, M., ... & Ouyang, W. (2025). Improving video generation with human feedback. arXiv preprint arXiv:2501.13918.
>
> [c] Xu, J., Huang, Y., Cheng, J., Yang, Y., Xu, J., Wang, Y., ... & Dong, Y. (2024). Visionreward: Fine-grained multi-dimensional human preference learning for image and video generation. arXiv preprint arXiv:2412.21059.

---

### Meta-Review · Area_Chair_VRiG · 2026-01-05

**Summary:**

This paper presents OneReward, a unified reinforcement learning framework for multi-task image editing using a single VLM judge. It outperforms commercial tools and removes the need for task-specific fine-tuning. However, the system relies entirely on the VLM being an accurate and unbiased judge. The large preference dataset required also makes the results difficult for others to reproduce. Authors argued the VLM shows high human agreement before the rebuttal. They later clarified that low-step sampling makes training efficient after the rebuttal. Overall, the paper is below the acceptance bar and the authors are encouraged to further improve the work.

**Reviewer Concerns:**

The rebuttal successfully addressed concerns about training efficiency and the accuracy of using one VLM instead of multiple experts. However, the lack of comparisons to recent RLHF baselines and missing ablation studies remain.

**Reviewer Scores:**

Reviewer PA5S might have increased their score if the multi-task conflict analysis was more thorough. Reviewer UTa8 likely would have maintained a low score due to the missing baseline comparisons. Reviewer Np2y would probably keep their positive rating because of the strong experimental results. Reviewer vrmd might have raised their score if the comparison with reward experts was in the main text.

---

### Decision · Program_Chairs · 2026-01-26

Reject